# Nanofibers in Ocular Drug Targeting and Tissue Engineering: Their Importance, Advantages, Advances, and Future Perspectives

**DOI:** 10.3390/pharmaceutics15041062

**Published:** 2023-03-25

**Authors:** Egemen Uzel, Meltem Ezgi Durgun, İmren Esentürk-Güzel, Sevgi Güngör, Yıldız Özsoy

**Affiliations:** 1Institute of Graduate Studies in Health Sciences, Istanbul University, Istanbul 34010, Türkiye; 2Department of Pharmaceutical Technology, Faculty of Pharmacy, Istanbul University, Istanbul 34126, Türkiye; 3Department of Pharmaceutical Technology, Faculty of Pharmacy, University of Health Sciences, Istanbul 34668, Türkiye

**Keywords:** nanofiber, electrospinning, non-electrospinning techniques, ocular drug delivery system, ocular tissue engineering

## Abstract

Nanofibers are frequently encountered in daily life as a modern material with a wide range of applications. The important advantages of production techniques, such as being easy, cost effective, and industrially applicable are important factors in the preference for nanofibers. Nanofibers, which have a broad scope of use in the field of health, are preferred both in drug delivery systems and tissue engineering. Due to the biocompatible materials used in their construction, they are also frequently preferred in ocular applications. The fact that they have a long drug release time as a drug delivery system and have been used in corneal tissue studies, which have been successfully developed in tissue engineering, stand out as important advantages of nanofibers. This review examines nanofibers, their production techniques and general information, nanofiber-based ocular drug delivery systems, and tissue engineering concepts in detail.

## 1. Introduction

Today, biopharmaceutics is increasingly using nanotechnology and nanoscience approaches [1]. Nanotechnology focuses on the creation, characterization, and application of nanomaterials. Nanomaterials are classified into four main categories: zero-dimensional (nanoparticles), one-dimensional (nanofibers), two-dimensional (nanofilms), and three-dimensional (polycrystals) nanomaterials. One-dimensional (1D) nanomaterials have received more attention because of their unique features, which include large surface area, small diameter, and high aspect ratio. The unique characteristics of 1D nanomaterials lead to applications in many areas, such as drug delivery, chemical and biological sensors, supercapacitors, and tissue engineering. As a result, the controlled synthesis of 1D nanomaterials is required for progress in new scientific research. Carbon nanotubes, nanobelts/nanowires, and nanofibers have been some of the 1D nanomaterials attracting much attention in nanotechnology applications [2,3].

Nanofibers have attracted attention among all of these alternatives due to their benefits. Nanofibers have recently been used in drug delivery applications due to their low preparation costs, ability to use a wide range of polymers, and ease of creation using different processes [4,5]. In addition, nanofibers have high porosities, large surface areas, adjustable mechanical properties, and changeable morphologies. These parameters can be customized based on the drug delivery conditions and requirements of various applications. Nanofibers are also a good candidate for tissue engineering due to their extracellular matrix (ECM)-like structure [6].

Nanofiber-based drug delivery systems have opened new opportunities in drug delivery. Nanofibers have been encapsulated with different therapeutic agents, such as small molecular medicines, and biological molecules, such as antibiotics, proteins, DNA, RNA, growth factors, and living cells [7,8]. Additionally, nanofibers can be used for topical, transdermal, and oral drug delivery, while short nanofibers can be injected locally into diseased areas. Furthermore, electrospun nanofibers provide significant benefits in terms of controlling drug release rates by modifying their composition (e.g., hydrophobic and hydrophilic materials) and structure (e.g., homogenous structure, core-sheath structure, and multilayered structure) [9,10,11]. Recently, it has been proven that targeted drug delivery systems can be prepared with stimuli-responsive nanofibers [12]. Thanks to their widespread use, nanofibers have become one of the most frequently studied subjects in the ocular field.

The eye, which has a unique anatomy, is the only vital organ in direct contact with the outside world. It contains many natural barriers to preserve its structural integrity. These anatomical barriers and physiology protect the eye from foreign substances and objects and act as rate-limiting steps for many drugs [13,14]. Therefore, developing ocular drug delivery systems and new-generation ocular tissues has always been a challenging area of interest to pharmaceutical technologists and tissue engineers. Nanofibers are one of the promising approaches to overcome these difficulties encountered in the ocular field, thanks to their features such as adjusting their porosity and fiber diameters, production with biocompatible materials and flexibility, and functionalization with different materials according to the purpose for use [15,16]. 

## 2. Nanofibers

Nanofibers are generally defined as filamentous structures less than 1 µm in diameter [17]. With the fiber diameter decreasing to nanometer size, the physical, chemical, and biological properties are highly developed, and nanofibers with a high surface area have emerged [18].

At the beginning of the areas where nanofibers are used the most, biomedical applications such as drug delivery systems, wound dressing, and tissue scaffold studies are coming (Figure 1). Due to the unique functionality and natural morphological properties of nanofibers, they are used as drug delivery systems. The diversity in the processability of nanofibers creates an advantage in terms of drug loading [19,20]. The regulation of the affinity between the polymers that form the structure of nanofibers and the drug also allows the adjustment of drug release profiles. The drug release profile varies according to the polymer properties forming the nanofiber, the production method, the different polymer combinations used, the surface modifications, and the behavior of the drug molecules [16].

Nanofibers have many advantages as a drug delivery system. They can be used for the controlled release of both hydrophilic and lipophilic active ingredients and biological molecules (such as antibiotics, proteins, etc.). Nanofiber morphology, porosity, and composition can also vary the drug release rate. Other advantages of nanofibers are that they have a high surface area/volume ratio and porosity, a structure similar to the extracellular matrix structure of tissue, and their production methods are less costly and simpler than many nanostructured drug delivery systems [21,22,23]. In addition, nanofibers can be designed as a targeted drug delivery system with thermoresponsive, pH-responsive, and electroresponsive properties [24,25,26,27]. Due to these properties, electrospun nanofibers, in particular, are getting more and more attention as a drug delivery system.

In the first trial of nanofibers used as a drug delivery system and produced by electrospinning, tetracycline hydrochloride was used as a model drug. Tetracycline-loaded sustained-release nanofibers were obtained with poly(lactic acid) (PLA) and poly(ethylene vinyl acetate) (PEVA) polymers [28]. In previous studies, nanofibers produced with many different drug molecules and polymers have been used in oral [29], transdermal [30], ocular [31], nasal [32], etc., applications.

### 2.1. Preparation Methods of Nanofibers

#### 2.1.1. Electrospinning

Electrospinning creates ultrafine fibers using an electric potential with a high voltage and low current. J. F. Cooley, who patented the technology as an “Apparatus for electrically distributing fibers” in 1902, was the first to observe an electrospinning process for such a purpose. The popularity of electrospinning grew at the end of the 20th century, when lots of new publications appeared and continue to appear, covering a wide range of applications for electrospun fibers, including drug delivery, wound healing, tissue engineering, textiles, sensors, cosmetics, and food packaging [33].

Electrospinning is a simple and successful method to produce nanofibers with diameters in the range of 10–1000 nanometers [34]. One of the main advantages of electrospinning is the high degree of control it provides over fiber morphology. This allows researchers to tailor the nanofiber properties to specific applications such as tissue engineering, drug delivery, and filtration. Additionally, electrospinning is a versatile technique that can be used with a wide variety of polymers, including natural and synthetic polymers as well as composites. This versatility allows the production of a wide variety of nanofibers with different properties. In addition, electrospun nanofibers have a high surface area-to-volume ratio, making them useful for a variety of applications [4,5,6,35,36]. Finally, electrospinning can be easily scaled up for mass production, making it suitable for commercial applications [37,38]. However, electrospinning also has some limitations. For example, the production rate is relatively low in conventional electrospinning devices, making it difficult to quickly produce large quantities of nanofibers. Additionally, electrospinning requires the use of high voltages, which can be hazardous, and the use of organic solvents, which can be harmful to the environment and human health. Finally, electrospinning can be costly due to the need for special equipment and the use of high-quality polymers [36,39,40]. In general, electrospinning is a powerful technique with many advantages, but these limitations must be considered when choosing a fabrication method for a particular application.

A polymer melt or solution can be used to manufacture nanofibers. Solution-based electrospinning has accounted for the majority of the research. The standard electrospinning system consists of four main parts: (1) a high electrical field, (2) a conductive capillary tube with a nozzle, (3) a syringe pump, and (4) a collector [41]. A high voltage (usually 10 to 30 kV) is applied to a polymer solution during electrospinning to create a charge on the droplet’s surface. The applied voltage causes a charge movement in the polymer liquid, which can stretch the pendant drop’s shape, which is usually spherical due to surface tension. When the charged polymer liquid’s electrostatic repulsion overcomes the surface tension, a conical shape known as Taylor’s cone is formed, and the jet initiation begins at the cone tip.

Interestingly, flow rate and applied voltage control the two forces that indirectly cause Taylor’s cone to form. As a result, the development of a stable jet is aided by a good balance between the two characteristics. If the polymer liquid has adequate cohesive force, a stable jet is ejected from Taylor’s cone, allowing the polymer chains to stretch and create a uniform filament. The solvent evaporates during this process, and dried polymer fibers are randomly distributed on the collector (Figure 2) [42]. The spinneret is essential in electrospinning since it allows for numerous configurations. The electrospinning process can be classified as needleless, single, coaxial, side-by-side, or triaxial electrospinning, depending on the spinneret [33,43] (Figure 2).

Numerous factors throughout the electrospinning process influence the nanofiber morphology. These variables are classified as follows: (i) solution parameters, (ii) process parameters, and (iii) environmental parameters. The solution parameters are the type of solvent, the molecular weight of the polymer, the solution concentration, the viscosity of the solution, the conductivity of the solution, the surface tension, the dipole moment, and the dielectric constant. The applied electric field, the distance between the needle tip and the collector, the flow rate, and so on are all process parameters. Ambient parameters that influence the electrospinning process include relative humidity and temperature [44]. The morphology of nanofibers is affected by all factors. All factors impact nanofiber morphology during electrospinning, and none act independently. As a result, to design a nanofiber with the appropriate structure and characteristics, the various parameters must be optimized [44,45]. The effects of the different working parameters on the fiber morphology are summarized in Table 1.

−Polymers for Electrospinning

Natural polymers (e.g., hyaluronic acid (HA), chitosan (CS), dextran (Dex), gelatin, and collagen), synthetic polymers (e.g., poly(lactic acid) (PLA), poly(lactic-co-glycolic acid) (PLGA), poly(-caprolactone) (PCL)), and their mixtures can all be used to create electrospun nanofibers [46]. Examples of synthetic and natural polymers and the solvents used in the formation of nanofibers for drug delivery are given in Table 2.

Nanofibers must fulfill particular requirements regarding mechanical properties, hydrophilicity, shape, and biocompatibility for each application. These characteristics are controlled by the chemical composition of the fiber, namely the polymer structure. The polymer structure has an impact on the loaded drug’s release rate and treatment duration, with three major factors at play: (i) polymer swelling in water, (ii) polymer affinity for the drug, and (iii) polymer degradation rate. In addition, the molecular weight of the polymer impacts some of the physical characteristics of the fiber, such as its thickness and physical stability. Because of the direct relationship between polymer molecular weight and solution viscosity at a constant concentration, molecular weight is also used to control the polymer concentration at which electrospinning may be conducted [63].

−Drug Loading Techniques in the Electrospinning Process

There are many different techniques for loading drugs in the electrospinning process. Some occur during electrospinning, while others occur with surface modification after electrospinning (post-electrospinning). In this review, electrospinning types used for drug loading will first be described and then post-electrospinning surface modification techniques will be discussed.

−Types of electrospinning

Although the most popular method is to blend electrospinning, there are many different types of electrospinning methods. Each type of electrospinning has its own advantages and disadvantages. In Table 3, the advantages and disadvantages of these different electrospinning types are briefly mentioned.

*Blend Electrospinning:* The blend electrospinning procedure in which drug/biomolecules are dispersed or dissolved in the polymer solution creates nanofibers with drug/biomolecules scattered throughout the fibers [70]. Despite its simplicity in comparison to other electrospinning techniques, the solvents used to disperse bioactive compounds can cause protein denaturation or loss of biological activity. Furthermore, the inherent charge of biomolecules can frequently cause migration on the jet surface, resulting in dispersion on the surface of nanofibers rather than encapsulation of biomaterials within the fibers. Surface dispersion has been linked to burst biomolecule release [64,70].

*Coaxial Electrospinning:* Coaxial electrospinning is an enhancement of conventional blend electrospinning such that two nozzles, rather than one, are connected to the high-voltage source. Two distinct solutions are put into each nozzle and pumped out to produce nanofibers with core-shell morphologies. Core-shell nanofibers with superior physicochemical and biological characteristics may be created using synthetic or natural polymers [71].

Electrospinning in a coaxial (or triaxial) direction is thought to be a viable approach for achieving sustained drug release from electrospun core-sheath nanofibers. It is formed by the simultaneous flow of a core and sheath solution from distinct capillaries. The production of core-sheath fibers by triaxial electrospinning, in which the electrospun fiber has a core, intermediate layer, and sheath, has recently been introduced. Multiple drugs may be put into the core-sheath fibers, and their release kinetics can be controlled using coaxial electrospinning. Coaxial spinning has a higher drug loading efficiency than blend spinning. More notably, the first burst release is observed to be lower in coaxially spun core-sheath nanofibers than in blend-spun fibers. After the hydrophilic component of the core dissolves, the core expands or dissolves, generating pores in the shell in core-sheath nanofibers. While drugs are being loaded into the core phase, the shell phase can act as an outer protective layer. In addition, incompatibilities can be solved. For example, hydrophilic drugs in the core phase can be integrated into hydrophobic polymers in the shell phase. Moreover, the shell phase can act as a physical barrier to provide sustained-release kinetics, and by loading the core and shell phases, two separate release patterns can be produced from a single delivery system [42].

*Emulsion Electrospinning:* In some sources, the emulsion electrospinning method is shown in the blend electrospinning category [72]. The configuration for emulsion electrospinning is similar to the setup for blend electrospinning. Emulsion electrospinning is the simultaneous spinning of two immiscible liquids to produce core-shell nanofibers. Active bioactive compounds and surfactants are allowed to create W/O emulsions before being blended with the polymer matrix solution [73]. The emulsion droplets are stretched into an elliptical shape during the spinning [71].

Moreover, the continuous-phase solvent evaporates quickly, resulting in a viscosity gradient and droplet enrichment in the axial area. Because of the viscosity gradient between the polymer matrix and the elliptical droplet, the core material settles inside the fiber matrix rather than on the polymer’s outer surface. This technology is simpler than coaxial electrospinning and provides sustained release of the loaded cargo components [71,74]. The applied voltage has a substantial effect on the diameters of the nanofibers. Higher applied voltages result in the formation of nanofibers with smaller diameters. Other parameters such as flow rate and spinning distance can also have an impact on fiber morphology. On the other hand, the interface tension between the organic and aqueous phases of the emulsion can harm the bioactive molecules placed within the emulsion electrospun fibers [71].

*Melt Electrospinning:* Melt electrospinning has risen in popularity in the electrospinning arena where toxicity and solvent accumulation were problems. Instead of using a solution, the polymer melt is used in this method, which is changed from liquid to solid after cooling to achieve the desired product, rather than using solvent evaporation. To produce high-quality fibers with uniform morphology but a wide range of diameters, the polymer-melt flowrate and homogeneous polymer-melt conditions must be controlled. Polymer blends and additives were used to reduce the average diameter of the fibers. The influence of melt temperature on the structure and function of drugs, proteins, and bioactive substances loaded in fibers can be substantial. The flowrate and melt viscosity of melt electrospinning could have an impact on the properties of the produced fibers. The surface wettability of melt electrospun fibers has been enhanced by forming hydroxyl, peroxyl, and N-containing functional groups using oxygen and ammonia plasma, respectively [71].

*Gas-Jet Electrospinning:* Gas-jet electrospinning is upgrading the conventional melt electrospinning process to produce nanofibers. In this method, polymer solutions are set up into a gas-jet device without the use of an electric field [71]. Additionally, the coaxial jet is surrounded by a tube feeding the heated gas, which can provide sufficient heat near the nozzle. Due to the coflowing gas jet, polymer solutions are exposed to additional stretching and shearing forces [75]. Thus, nanofibers produced in this method become thinner and more homogeneous demonstrating that polylactic acid nanofibers have thinner diameters when the gas flow rate is increased [76].

*Side-by-side Electrospinning:* Side-by-side electrospinning is a process in which two liquids are applied from separate capillaries. In some sources, it is indicated as a coaxial electrospinning type. However, they are placed adjacent to each other rather than two polymer solutions nested inside each other as in the coaxial process [77]. For example, Yang et al. used a side-by-side electrospinning method to create PVP and ethylcellulose nanofibers with a synergistic release of ciprofloxacin and silver nanoparticles, both of which have antibacterial activity. With this Janus approach, the fibers were able to burst and release ciprofloxacin within 30 min. The antibacterial effect was then maintained for 72 h by the sustained release of silver nanoparticles, significantly inhibiting bacterial growth. Because of this characteristic, this scaffold is a promising method for preventing infections throughout the wound-healing process [78].

−Post-electrospinning surface modification techniques

Several methods have been used to immobilize drugs on the surfaces of electrospun polymeric nanofibers. These can be listed as follows: (i) physical immobilization by simple physical surface adsorption, (ii) layer-by-layer assembly, or (iii) chemical immobilization methods. These methods can provide stronger and more stable drug immobilization on the nanofiber’s surface [79].

*Physical Surface Adsorption:* Physical surface adsorption is the easiest way to incorporate bioactive substances into membranes and does not rely on chemical treatment. Physical absorption refers to the immersion of electrospun fibers in a liquid solution, where the internal substances tend to adhere to the surface of the scaffolds through electrostatic interaction. Weak nonspecific intermolecular interactions (such as those found in electrostatic interactions, hydrogen bonds, hydrophobic interactions, and Van der Waals forces) are generally established between the surface and peptide sequences [79,80]. Casper et al. created electrospun PEG nanofibers that were functionalized with low molecular weight heparin, a highly sulfated glycosaminoglycan that binds growth factors (GF) for drug delivery and wound healing. The results show that this functionalization technique allows essential fibroblast growth factor (bFGF) to bind to the surface of PEG nanofibers [81]. However, due to the uncontrolled release of drugs, this approach is rarely used for loading proteins or nucleic acids. For example, Wang and colleagues constructed electrospun PLGA/hydroxyapatite (PLGA/HAp) microfibers encapsulated with bone morphogenetic protein-2 (BMP-2). Although the release of adsorbed BMP-2 took more than two weeks, 75% of the proteins were released within the first five days. There was also a distinct burst release phenomenon for gene transfer, and the released gene was rapidly degraded [82].

*Layer-By-Layer Assembly (LbL):* Generally, layer-by-layer assembly is a sophisticated method of alternating the adsorption of materials onto a surface utilizing opposite interactions, with one layer of material deposited at a time. LbL assembly is a cyclical process that involves the alternating deposition of polymers with opposite charges on their surfaces to build a covering of polyelectrolyte multilayers (PEMs) or a free-standing film [83]. The depositing process can be continued indefinitely until a multilayer film of the required thickness is produced. Electrostatic attraction is the primary driving factor in the assembly process, but hydrogen bonding, hydrophobic, covalent, and biological interactions can also be important. The film’s composition, morphology, and structure may be accurately controlled by this build-up [84]. Xine et al. created cellulose nanofiber mats coated with silk fibroin (SF) (negatively charged) and lysozyme (positively charged). The in vitro and in vivo studies revealed that the mats can enhance wound healing as well as wound infection prevention [85].

*Chemical Immobilization:* Chemical immobilization refers to the covalent bond that binds agents on the surface of fibers. External enzymes can control the release of agents. The surface characteristics of fibrous membranes can be modified using this technique. Because of the complexity of the immobilizing agents at the membrane surface, covalent immobilization is not a widely used method of loading bioactive compounds [72]. For instance, to immobilize bioactive molecules on the surface of nanofibers for wound healing, the chemical immobilization of primary amine and carboxylate groups has been widely used [86].

#### 2.1.2. Non-Electrospinning Methods

−Interfacial Polymerization

In this process, two different monomers that can be dissolved in two separate phases (such as oil and water) are used. After dissolving the two monomers, they will polymerize at the interface of the emulsion droplet. For example, after dissolving diamine in the water phase, the diacid chloride (oil soluble) is added to the solution and reacts with the first monomer at the interface forming the wall material. Nanofibers can be created by using this method due to homogeneously nucleated growth. Different polymers may also be produced by selecting other monomers; however, most publications refer to polyamide membranes [87].

−Drawing

Another process for producing fibers is drawing. It is similar to spinning. The fact that this process requires just a sharp tip or a micropipette is considered its main advantage. A sharp tip is used in this procedure to draw a droplet of a previously placed polymer solution as liquid fibers. Then, the solvent is evaporated due to the high surface area, allowing the liquid fibers to solidify. Hollow glass micropipettes can be used rather than the sharp tip with a continuous dose of the polymer to avoid the volume shrinking problem that limits the continuous drawing of the fibers and affects their diameter. The micropipette is gently pulled from the liquid and moved at a low speed after being dipped into the droplet using a micromanipulator. Consequently, nanofibers are pulled and deposited on the surface by touching it with the end of the micropipette. To produce nanofiber, this process was repeated several times on each droplet. It can also be used to create continuous nanofibers. Furthermore, exact control of essential drawing parameters such as drawing speed and viscosity can be achieved, allowing for repeatability and control over the dimensions of the produced fibers. Although this method is simple and the dimensions of the fibers can be controlled, it is limited to the laboratory scale because nanofibers are produced one at a time. Moreover, it is a discontinuous process with low performance. To withstand the stress produced by the pulling, only viscoelastic material can be used in this process. In addition, depending on the orifice size, fibers with diameters larger than 100 nm can be produced [87,88].

−Template Synthesis

In this process, using chemical or electrochemical oxidative polymerization, polymeric nanofibers can be produced using a nonporous membrane with several cylindrical pores [87].

One of the features of this technology is the ability to produce nanofibers with varying diameters by using different templates [87]. It implies using a template or mold to achieve a desired material or structure and produce nanofibers. A metal oxide membrane is referred to in the template. In this process, nanofibers are produced by passing polymer solution through pores of nanoscale diameter while applying water pressure on one side, causing extrusion of the polymer and the production of fibers upon contact with the solidifying solution. This technique cannot create nanofibers with long fiber lengths; only a few micrometers are obtained, and the membrane’s pore size controls the diameter of these fibers [88].

−Phase Separation

In phase separation, a polymer is first blended with a solvent before being gelated. The solvent phase is then extracted, leaving the other remaining phase. This system’s main mechanism is separating phases due to physical inconsistency. Some sources describe a thorough process in five steps: (i) polymer dissolving in a solvent at normal temperature or increased temperature; (ii) gelation is the most challenging part of controlling the nanofiber morphology (porosity); (iii) the duration of gelation varied with polymer concentration and gelation temperature; (iv) water extraction of the solvent from the gel; and (v) freezing and freeze-drying under vacuum [89].

It is the minimum equipment needed for the process. It can directly create a nanofiber matrix where mechanical characteristics can be modified by changing the polymer concentration. This process can produce long continuous fibers; however, not all polymers can undergo phase separation and create nanofibers since gelation capacity is required, which limits the use of the phase separation method [87]. Only a few polymers, such as polylactide (PLA) and polyglycolide, have been turned into nanofibers using the phase separation process so far [88].

−Self-Assembly

Self-assembly is a bottom-up nanomaterial production approach in which molecules organize and arrange themselves into patterns or structures via noncovalent forces such as hydrogen bonding, hydrophobic forces, and electrostatic interactions. It is an effective technique for creating very small nanofibers with lengths of several micrometers by forming supramolecular hydrogels via weak interactions such as hydrogen bonding and hydrophobic interactions [90]. The basic process is based on intermolecular interactions that bring tiny units together; the structure of the smaller units of molecules dictates the overall structure of the nanofiber [91]. The major disadvantage of the process is that it is a complex, long, and highly elaborate procedure with low production and a lack of precise control over the fiber dimensions. Furthermore, this process is limited to the production of nanofibers from small active molecules that may self-assemble or be stimulated by an external stimulus [87,90,91].

−Freeze-Drying (FD)

Freeze-drying is a process that involves three main steps: first, the solution is frozen at a low temperature (−70 °C to −80 °C). Then, the frozen sample is placed in a chamber where the pressure is reduced to a few millibars through a partial vacuum. This is known as the primary drying process, which removes ice from the material by direct sublimation. In the next step, most of the unfrozen water in the material is removed by desorption in a secondary drying process. In recent years, freeze-drying has been extensively studied for the production of three-dimensional porous scaffolds [92,93]. In some studies, as a result of the optimization of the freeze-drying method, the diameters of this scaffold are reduced to nanosize, so researchers define this structure as nanofiber [94,95,96,97]. For example, Ma et al. used the freeze-drying method to create chitosan/sodium hyaluronate polyelectrolyte complex fibers. In vitro tests showed that the resulting fibers had good compatibility and no cytotoxicity [98]. In another study, Lee et al. obtained the nanofiber structure of polypyrrole nanoparticles using the freeze-drying method [99].

Moreover, freeze-drying contributes to the production of nanofibers by applying it together with other different methods as well as being used in the phase separation method, as mentioned before [89]. For instance, Lin et al. synthesized ultrafine porous boron nitride nanofiber mats in a two-step production method including freeze-drying and pyrolysis methods. The resulting nanofiber mats have a high surface area and porosity [100]. Many studies also use freeze-drying in combination with the electrospinning method. By combining these two methods, a material with a high surface area and small fiber diameters of electrospun materials can be created, as well as the structural integrity and porosity of freeze-dried samples [101,102]. For example, Tang et al. combined electrospinning and freeze-drying techniques to fabricate membranes, which were found to have increased absorption capacity with increasing porosity [103].

Freeze-drying (FD) has several significant advantages over other methods. It can create porous structures with controllable sizes directly from polymers such as chitin without the requirement for structure-directing additives or pretreatments, which other techniques such as self-assembly and electrospinning cannot.

Furthermore, the freeze-drying technique does not require a high temperature or further leaching step, and the use of water and ice crystals rather than an organic solvent in the scaffold production process makes this process more appropriate for biomedical applications. As a result, it has attracted increased interest in the production of nanofibers. Even though the freeze-drying process has various advantages, it is still challenging to construct scaffolds with hierarchical structures (e.g., vascularized systems) using this method. This method’s nanofiber mats can be used as drug delivery systems, as precursors for highly porous carbon nanofibers, or as templates for manufacturing inorganic fibers [87,104].

−Rotary or Centrifugal Jet Spinning (RJS/CJS)

Rotary or Centrifugal jet spinning (RJS/CJS) (also called forcespinning or rotational jet spinning) is a novel method for the fabrication of nanofibers with controlled size, morphology, and orientation. RJS/CJS differs from electrospinning in that it uses centrifugal force instead of an electric field to produce the spinning process [105,106]. This technique involves the use of a rotating spinneret that ejects a polymer solution or melts through a small orifice (nozzle). At a certain rotational speed, the centrifugal force becomes more powerful than the surface tension of the liquid being rotated, forming liquid jets from the rotating head’s nozzles. These jets are then elongated by the combination of centrifugal force and air friction, ultimately creating nanofibers. The fibers can be collected on a rotating mandrel or a stationary collector, resulting in aligned or randomly oriented fibers, respectively [90,107]. The process of creating nanofibers through RJS/CJS involves controlling the centrifugal force, viscoelasticity, and mass transfer of the solution to manipulate the solution filament into thin nanofibers. The diameter of the resulting nanofibers is greatly influenced by the spinning solution’s elasticity and the solvent’s evaporation rate [108]. In addition, the RJS/CJS process for nanofibers is influenced by multiple factors, including the speed at which the heated structure rotates, the design of the nozzle, the collection system used (such as the distance between the collector and spinning apparatus), and the temperature. These factors affect the shape and appearance of the resulting nanofibers [109,110].

The RJS/CJS technique can be divided into three types, namely melt RJS/CJS, immersion RJS/CJS, and nozzleless RJS/CJS. In melt RJS/CJS, molten polymer is used in the RJS apparatus to create micro and nanofibers without the need for solvents. High temperatures are used to melt the polymer, and additives such as viscosity-reducing agents or plasticizers are used to improve fluidity and reduce fiber diameters [111]. Immersion RJS/CJS involves dropping the extruded polymer into a solution for solidification and/or crosslinking, which minimizes extrusion breakage and bearing in the fibers by reducing surface tension [111,112]. Nozzleless RJS/CJS, on the other hand, does not use a needle for polymer extrusion and instead creates “fingers” by pulling out of a lid-disk gap due to Rayleigh–Taylor instability. This method is an alternative to nozzle RJS systems that can clog when highly viscous polymers are used [111].

Nanofibers produced by the RJS/CJS method have been used with increasing interest in drug delivery systems and tissue engineering studies in recent years. For example, carvedilol was incorporated into nanofibers using an RJS/CRS method, and the nanofibers were then formulated into dispersible tablets. Dissolution studies showed that the drug released quickly and completely from the nanofiber-based tablets, regardless of pH [113]. Another study used PCL and gelatin as model polymers in C-spinning to produce highly aligned ultrafine fibers with smooth surfaces. The resulting mat showed improved hydrophilicity, porosity, and mechanical properties, making it suitable for tissue engineering applications. In vitro and in vivo experiments confirmed the biocompatibility of the scaffolds, which can be used as a wound dressing material [114].

Badrossamay, M. R. et al. engineered nanofiber constructs with robust fiber alignment from collagen, gelatin, and PCL blended using the RJS/CJS method and compared them to electrospun fibers. The RJS/CJS-spun fibers had a higher production rate and retained more protein content on the surface. The authors demonstrated the biofunctionality of RJS/CJS scaffold fibers by testing their ability to support cell growth and maturation with different cell types. The hybrid nanofiber constructs fabricated by RJS/CJS have the potential to be used as a scaffold material for a wide variety of biological tissues and organs as an alternative to electrospinning [115].

By changing the spinneret type, nanofibers of different structures can be obtained in the RJS/CJS method, just as in the electrospinning method. Khang, A. et al. obtained anisotropic two-phase Janus-type nanofiber scaffolding meshes using PCL and gelatin polymers by the centrifugal jet spinning method. The resulting scaffold meshes exhibited a variety of mechanical properties and surface chemistries that are potentially useful in tissue engineering applications. The study provides evidence for the ability of this method to precisely control the orientation of the fiber web and the distribution of different materials within the scaffold [116].

In conclusion, the centrifugal spinning method has become quite popular in drug delivery systems and tissue engineering studies in recent years. The main reasons for this are its high productivity and low solvent usage, as well as being a relatively low-cost nanofiber production method that does not require high voltages [105]. Additionally, successful scale-up studies have been conducted [117]. However, the main disadvantage of centrifugal spinning is that the quality and productivity of fibers are highly affected by material properties and nozzle design [111].

## 3. Nanofibers as an Ocular System

### 3.1. Anatomy and Physiological Barriers of the Eye

Anatomically, the eyes consist of two segments, anterior and posterior. The lens, which is a transparent structure responsible for refracting the light coming into the eye, is the border that separates these two parts [118]. The anterior segment includes the cornea, iris, ciliary body, conjunctiva, and anterior surface of the sclera. The choroid, retina, optic nerves, and the posterior surface of the sclera are located in the posterior segment. In both the anterior and posterior segments, the interstitial spaces are filled with aqueous humor and vitreous humor, respectively [13].

The anatomical structure of the eye can be classified independently of its segments, as well as according to the structural features of the layers that make it up. These layers are from outside to inside: *Tunica fibrosa oculi* (fibrous layer), *tunica vasculosa oculi* (vascular layer), and *tunica interna oculi* (neural layer). *Tunica fibrosa oculi*, the fibrous layer of the eye, consists of the cornea, conjunctiva, and sclera. This layer, which does not contain lymph and blood vessels, surrounds the eyeball from the outside [119]. The cornea is a thin and transparent tissue consisting of five layers (corneal epithelium, *Bowman’s* membrane, stroma, *Descemet’s* membrane, and the endothelial layer). The conjunctiva is a mucosal membrane [13]. The sclera is a fibrous tissue consisting of five layers (Tenon’s capsule, episclera, scleral spur, limbus, and posterior sclera) [120], just like the cornea. The *tunica vasculosa oculi*, located just below the tunica fibrosa oculi, is also known as the uvea. This layer consists of the choroid, ciliary body, and iris. Thanks to the vessels it contains, the blood circulation in the eye tissues and the production of aqueous humor are carried out by the tissues in this layer [13,119]. The *tunica interna oculi* is the inner layer where the light is transformed into a neural impulse and transmitted to the brain [121].

Since the eye is a vital organ in contact with the external environment, it must maintain its integrity. For this reason, it contains natural anatomical and physiological barriers. All ocular tissues, vessels, and fluids are the natural barriers of the eye. These barriers are the tear film, cornea, conjunctiva, sclera, blood–aqueous humor (iris–ciliary body), lens, and blood–retina barrier [122,123,124]. These barriers, whose main task is protecting the eye from foreign fluids and objects, also act as drug rate-limiting steps. They reduce the penetration of drugs into ocular tissues and their bioavailability. Since each of these barriers has different structures from the others, they also perform rate-restriction functions on drugs with different properties.

The tear film covers the surface of the cornea and conjunctiva. It is the primary protective structure against ocular problems of chemical, mechanical, bacterial, or viral origin. Likewise, it forms the largest barrier among drugs applied topically to the eye [120,125]. Topically applied drugs are usually applied to an area called the *cul-de-sac* with a volume of 7–10 µL [126]. However, the application dose of any drug is 20–50 µL on average [127]. Due to this sudden increase in the volume of the area, topical drugs such as drops and emulsions applied topically are removed from the eye by nasolacrimal drainage and added to the systemic circulation [123,128]. Around 80% of a topically applied conventional drug is eliminated due to the tear film and nasolacrimal drainage [129].

The cornea acts as an important barrier to conventional topical drugs in the anterior segment of the eye [130]. The epithelium, stroma, and endothelium, which are parts of the cornea, are tissue formations of different polarities. Depending on the arrangement of the cells in the epithelial layer, there is a high shunt resistance and is considered a tight tissue [122]. These tight junctions give the epithelium a lipophilic character. Stroma, conversely, has a highly hydrophilic character due to the collagen fibers in its structure. Therefore, while the epithelium prevents the passage of hydrophilic drugs, the stroma acts as a barrier to the passage of lipophilic drugs. In the endothelium, the innermost layer of the cornea, molecules of up to 70 kDa can pass through passive transport [123,124,130].

The conjunctiva acts as a barrier through the numerous capillaries and lymphatic structures in its structure. It dilutes applied drugs with the blood or lymphatic circulation. There are epithelial cells in the conjunctiva as well as in the cornea. These epithelial cells have a shunt resistance, although not as much as in the corneal epithelium. Points of this resistance are considered tight structures and act as barriers to drugs. The conjunctiva acts as the main barrier for drugs not administered via the corneal route. Compared to the cornea, the conjunctiva is a hydrophilic tissue and more suitable for passing large molecules. For this reason, the direct conjunctival application of ocular drugs is being studied by reducing corneal application to increase the absorption of larger bio-organic molecules such as proteins and peptides. However, considering that most of the drugs used in the clinic are lipophilic and small molecules, it is seen that corneal application will not lose its importance [120,122,130].

The sclera has a hydrophilic character due to the collagen fibers in its structure. For this reason, it acts as a barrier for lipophilic drugs. However, small-sized lipophilic drugs also show scleral permeability similar to hydrophilic drugs. In addition, the size of the molecules and the positive charge are other parameters that reduce the passage through the sclera [124,131].

The iris and ciliary body, the tissues of the tunica vasculosa bulbi form the blood–aqueous barrier (BAB). This barrier is also known as the anterior blood–eye barrier [124]. The ciliary body produces aqueous humor, which stabilizes intraocular pressure. On the other hand, the dense capillaries in this layer contain tight junctions. These tight junctions, the epithelial cells of the iris, and the barrier formed by aqueous humor are highly restrictive for hydrophilic drugs. Small molecules and lipophilic drugs can easily pass the BAB and enter the systemic circulation via the uvea. An inflammation that may occur in this barrier disrupts the integrity of the barrier. This causes the amount of drug passing to the anterior region of the eye to not be controlled [122,124,131,132,133].

The lens is the eye’s tissue, consisting of 65% water and the remainder predominantly proteins. It is distinctly different from other eye tissues with its protein content. The lens is a tissue in which active and passive transport, depending on Na^+^, are effective. It mainly affects the passage of drug molecules for this reason [122,131].

The blood–retinal barrier (BRB) is formed by the combination of several factors. Primarily, retinal pigment epithelial cells have *Na^+^*, *K^+^-ATPase* pumps that ensure the balance of Na^+^ and K^+^ ions in the eye. On the other hand, the capillaries in their structure have tight connection points. The passage of drug molecules through these tight junctions is the second rate-limiting step. At the same time, molecules that can pass through these capillaries quickly enter the systemic circulation, which reduces the number of drugs in the target area, thus reducing the bioavailability. Finally, the ganglion cells in the retina are nerve cells and form a barrier similar to the cerebrospinal barrier. The BRB acts as a barrier for the passage of proteinaceous and small hydrophilic molecules while allowing the passage of lipophilic molecules. However, these lipophilic molecules are also rapidly distributed and eliminated due to intense systemic circulation [120,122,124,131].

Increasing the ocular bioavailability of a drug to be applied to the eye can only be achieved by overcoming all of these barriers and delivering the drug to the target tissue. Two ways are followed for this purpose. One of them is the application of the drug to the eye in different routes. These routes of administration are classified as topical, periocular (subconjunctival, transscleral, and intravitreal), and systemic [120,124]. Although these different routes, which are actively used today, seem to have solved the problem of low ocular bioavailability, they contain risks as they can cause serious complications such as blindness. For this reason, the second way to increase ocular bioavailability is the development of drug delivery systems [118].

### 3.2. Drug Delivery System

Conventional dosage forms applied to the eye are solutions, suspensions, ointments, and gels [134,135]. Today, 90% of the drugs on the market fall into this group. These dosage forms usually target the anterior segment of the eye and may have little effect on a possible disease in the posterior segment [136]. On the other hand, they have very low bioavailability due to ocular barriers. Due to their low bioavailability, their dosing frequency during the day is high, which reduces patient compliance. In addition, the fact that gels and ointments cause blurred vision is another factor that reduces patient compliance [137]. To prevent all of these negativities, drug delivery systems targeting certain eye tissues are being developed.

Drug delivery systems are produced by using a suitable polymer/surfactant/lipid, have micro or nano size, and are developed to deliver the loaded drug to the target tissue [138]. Since they target the drug to the tissue where the effect is desired, they allow the use of active pharmaceutical ingredients (API) at lower doses. Thus, a decrease in the rate of side effects may occur. In addition, drug releases can be extended thanks to the polymer/surfactant/lipid used in their preparation. Thus, the dosing frequency can be reduced and patient compliance can be increased [139,140,141]. There are different delivery systems developed for ocular drug targeting. Micro and nanoparticles, micro and nanoemulsions, nanosuspensions, micelles, liposomes, solid lipid nanoparticles, dendrimers, cubosomes, discosomes, niosomes, spanlastics, bilosomes, hydrogels, implants, inserts, lenses, and nanofibers are among the carrier systems developed [118]. While some of these carrier systems have turned into commercial products, some have remained in experimental studies. The most well-known commercial products are Ocusert^®^ (an ocular insert containing pilocarpine) [142], Xelpros^®^ (a pilocarpine-containing micellar system) [143], Modusik-A Ofteno^®^, Papilock Mini^®^, and Cequa^®^ (which are micellar systems containing cyclosporine A) [144,145].

Nanofibers have also been used in ocular drug targeting since the first years they were designed as drug delivery systems. Thanks to the materials used in their preparation, they increase the penetration and contact time of drugs with ocular tissues and show high biocompatibility. In addition, their extended-release profile also reduces the dosing frequency [146,147]. However, the features that make nanofibers distinctly superior to other drug delivery systems are that they have a large surface area, high porosity, easily adjustable diameters, and can be combined with other drug delivery systems [15]. Having a large surface area enables the nanofibers to be loaded with more drugs than other drug delivery systems. These features, coupled with extended drug release profiles, allow for further reductions in dosing frequencies. For this reason, nanofibers stand out as a promising approach to the medical treatment of chronic ocular diseases. The fact that porous structures can be produced to be similar to ocular tissues further increases their biocompatibility compared to other carrier systems [148,149,150]. One of the most important features of drug delivery systems is that they are micro or nanosized. The diameter size of nanofibers could be adjusted much more easily than other carrier systems. Thus, the permeation of the drug with ocular tissues can be increased. The studies found that the carrier systems should be smaller than 100 nm to overcome the corneal barriers [151], and the scleral pore openings were between 20–80 nm [13]. Nanofibers can be easily produced in the appropriate diameter according to the target tissue. It also provides an important advantage in that it can be used in combination with other drug delivery systems (nanoparticle, dendrimer, etc.). In recent years, studies in the hydrogel form of nanofibers, which are generally converted into insert or implant form, have also been carried out. Although the drug has been proven to increase penetration and permeation with ocular tissues, it can cause blurred vision, irritation, and watery discharge. Despite these problems, nanofibers may be a viable option for a variety of ocular drug delivery systems [152]. To avoid these side effects and maximize the benefits of using nanofibers for ocular drug delivery, nanofiber properties such as size, surface charge, and polymer type and composition can be optimized, which can significantly affect their interaction with ocular tissues. By optimizing these properties, researchers can minimize the negative effects of nanofiber-based drug delivery systems while maintaining their effectiveness. Thus, the irritation potential of formulations can be reduced by reducing interactions with ocular tissues. In addition, nanofibers can be combined with other delivery systems such as hydrogels or liposomes to increase their performance and reduce their adverse effects. For example, nanofibers prepared with hydrogelizing polymers can provide a protective barrier by minimizing irritation and other side effects. In Table 4, nanofibers developed as an ocular drug delivery system are summarized (in alphabetical order).

### 3.3. Tissue Engineering

Many tissues in the living body cannot regenerate after injury. Even with surgical intervention on these damaged tissues, the tissue cannot regain its former form and return to the former quality of life of the individual [179]. One of the methods developed to solve this problem is tissue engineering. Tissue engineering, also known as regenerative medicine, is the science that deals with the development or production of therapeutic stem cells, tissues, and artificial organs [180]. Tissue engineering is widely used in bone and nerve regeneration, cartilage repair, and cardiovascular and ocular tissue development. Although it may seem simple and feasible, it is a multidisciplinary field of study and contains various difficulties in each step of obtaining tissue from stem cells [109].

The extracellular matrix (ECM) is a heterogeneous, binding network of fibrous glycoproteins with micro- and nanosized pores [181]. It provides the physical scaffold and mechanical stability required for tissue morphogenesis and homeostasis. It is aimed at developing scaffolds similar to ECMs in tissue engineering [148,149,150]. Electrospinning is the widely used method in producing scaffolds produced using biodegradable materials to obtain micro or nanosized fibrous tissue.

The eye has a unique structure and is rich in epithelial cells and neural networks. However, the structural integrity of this organ, which is in contact with the outer surface, may be impaired due to chemical, radiation and burn injuries, and infections caused by contact lenses. Unfortunately, these structural problems can be permanent. For example, the cornea, the outermost transparent layer of the eye, has two main functions. The first is to protect the structures inside the eye, and the second is to refract the light coming from outside and focus it clearly on the retina [182]. Good vision largely depends on corneal epithelial regeneration by limbal epithelial cells [183]. However, structural problems due to environmental factors may lead to limbal stem cell deficiency, and in this case, the act of seeing may not be performed. Therefore, the development of ocular tissues is an important field of study for tissue engineering. It is a great advantage that many biodegradable polymers used in the production of scaffolds are also biocompatible with ocular tissues. Table 5 summarizes the ocular tissue engineering studies with nanofibers.

When the studies are examined, it is generally emphasized that nanofibers produced with different methods and different materials in ocular tissue engineering studies are highly biocompatible with eye tissue and suitable for adhesion and proliferation for different types of cells [184,185]. In addition, successful results have been obtained from anti-inflammatory gene expression studies [186,187]. In other studies, it has been concluded that the prepared nanofibers have better mechanical properties than the amniotic membrane [188]. In some in vivo studies, it has also been observed that cell-cultured nanofibers provide re-epithelialization in the eye [189,190].

**Table 5 pharmaceutics-15-01062-t005:** Nanofiber-based ocular tissue engineering studies.

Tissue	Polymer	Comments	References
Limbal stem cell	PHBV	Electrospun nanofiber scaffold has favorable mechanical, physical, and chemical properties. Biocompatible nanofiber scaffolds. Similar transparency compared to the amniotic membrane (AM) in wet scaffolds. Nanofiber scaffold suitable for cell adhesion. The successful proliferation of limbal stem cells (LSCs) on nanofiber scaffolds in in vitro studies.	[184]
Limbal stem cell	PCL	Biocompatible nanofiber scaffolds. Limbal epithelial cells (LECs) can be easily attached to the nanofiber scaffold. LECs are active and proliferate on the nanofiber scaffold. Immunofluorescence (IF) staining and reverse transcriptase polymerase chain reaction (RT–PCR) results show the same expression profile of LECs on the scaffold and AM. Cells infiltrate the nanofibers and form a viable three-dimensional (3D) corneal epithelium.	[185]
Retinal pigment and corneal epithelial cells	PCL	Two different nanofiber scaffolds with two different diameters were formed using the electrospinning method (527 ± 184 nm and 1309 ± 116 nm, respectively). Retinal pigment epithelial cells (ARPE-19) and corneal epithelial cells (HCE-T) cultured on the nanofiber scaffold are both metabolically active on the two nanofiber scaffolds. However, ARPE-19 shows better adhesion and metabolic activity in scaffolds with a larger diameter. HCE-T cells cultured on ~500 nm nanofiber show higher proliferation, differentiation, and lower apoptotic markers. However, HCE-T cells on ~1300 nm nanofiber show higher stem cell expression. ARPE-19 cells cultured on ~500 nm nanofiber have a higher level of secretion of VEGF-A (Vascular endothelial growth factor A) compared to larger diameter nanofiber.	[191]
Conjunctival epithelial cells	SF/PLCL	Hydrophilic, smooth, homogeneous electrospun nanofiber scaffolds. Biocompatible nanofiber scaffolds. In in vitro studies, rabbit conjunctival epithelial cells (rCjECs) and goblet cells successfully adhere and proliferate on the nanofiber scaffold. Successful Cjec gene expression from the nanofiber scaffold, on one hand, decreases the expression of inflammatory mediators. In vivo studies show that CjECs become more stratified, while the nanofiber scaffold structure degrades.	[192]
Retinal ganglion cells	PPy-G/PLGA	Biocompatible electrospun nanofiber. In in vitro studies, retinal ganglion cells (RGCs) are significantly active and neurite outgrowths occur on the nanofiber. Electrical stimulation to nanofibers provides an anti-aging effect on RGCs, improving cell length with 137% elongation.	[193]
Ciliary pigment epithelial cells	RADA-16-I peptide	Nanofiber scaffolds produced by the self-assembly method. Ciliary pigment epithelial stem cells (CPE-NS) were successfully encapsulated in a nanofiber scaffold. Encapsulated CPE-NS have similar activity and proliferation compared to normal cell culture. Encapsulated cells express neural progenitor markers through ambient (medium) conditions. Encapsulated cells differentiate into the retinal neuronal direction.	[194]
Limbal stem cell	dAM/PCL	PCL nanofiber prepared by the electrospinning method. The composite membrane is prepared by conjugation of PCL nanofiber and dAM. Compared to dAM, the composite membrane has improved integrity and mechanical properties. In in vivo studies, the composite membrane is as immunomodulatory as dAM. The composite membrane is more suitable to support LSC survival, retention, transport, and proliferation compared to dAM alone. In in vivo studies, the composite membrane promotes eye re-epithelialization and reduces inflammation and neovascularization.	[189]
Limbal stem cell	PCL	Biocompatible electrospun nanofiber. Nanofiber scaffold suitable for cell adhesion. The successful proliferation of LSCs on nanofiber scaffolds in in vitro studies. Nanofibers take the form of viable 3D corneal epithelium from which cells can infiltrate. No significant difference in the expression profile of LECs grown on nanofibrous scaffold compared to those cultured in human AM.	[195]
Limbal stem cell	PLA	Mesenchymal stromal/stem cells (BM–MSCs) from bone marrow, mesenchymal stromal/stem cells (Ad–MSCs) from adipose tissue, and LSCs successfully grow and proliferate on the nanofiber. Cells on the nanofiber scaffold adequately migrate to the ocular surface. In in vivo studies, cell-cultured nanofiber scaffolds show improvement in re-epithelialization, neovascularization, and corneal thickness properties in injured rabbit eyes.	[190]
Limbal stem cell	Carbodiimide cross-linked AM	As a result of crosslinking of AMs with carbodiimide, a triple helix molecular AM collagen structure is formed and a nanofiber scaffold is obtained. The helical structure becomes a more random globular state as the crosslinking time of AMs increases. As the crosslinking time of AMs increases, the nanofiber diameter becomes larger. Compared to AM, cross-linked AMs show increased water content, light transmittance, and resistance to enzymatic degradation. LECs show growth and gene expression on cross-linked AM nanofiber scaffold in in vitro studies.	[196]
Limbal stem cell	PCL/PLA/PLGA/dAM	Individually PCL, PLA, and PLGA electrospun nanofibers conjugate with dAM to form composite membranes. Compared to dAM, the composite membranes have improved integrity and mechanical properties. Composite membranes show bioactivity similar to dAM in terms of LSC adhesion, growth, and proliferation. Anti-inflammatory gene expression was similar to dAM from composite membranes.	[186]
Llimbal stem cell	PCL	Surface-modified nanofiber scaffold formation by applying helium–oxygen discharge to electrospun PCL nanofibers. No significant morphological difference between surface-modified and non-surface-modified nanofibers. The surface-modified nanofiber scaffold has improved wettability and transparency. Both nanofiber scaffolds (surface-modified and non-surface-modified) are biocompatible. Cell adhesion and proliferation are better in surface-modified nanofiber. Similar gene expression is seen in both nanofiber scaffolds.	[197]
Limbal stem cell	Silk	Silk nanofiber scaffold produced by electrospinning method. Biocompatible nanofiber scaffolds. LSCs grow and proliferate successfully on nanofiber. LSCs infiltrate the nanofibers and form a viable three-dimensional (3D) corneal epithelium. As a result of in vitro tests, the gene expression profile of LCSs on nanofiber is similar compared to AM.	[198]
Retinal progenitor cells	LPG/DPG/RPG	Nanofibrous hydrogel structures produced by the self-assembly method. DPG and LPG hydrogels form in a helical structure while RPG does not have a helical form. Nontoxic nanofiber hydrogels. Compared to LPG nanofibers, DPG nanofibers promote better neuronal differentiation, migration, and synapse formation of retinal progenitor cells (RPCs).	[199]
Limbal and mesenchymal stem cells	Copolymer PA6/12	Biocompatible electrospun nanofiber scaffolds. LSCs, MSCs, and corneal epithelial and endothelial cell lines show similar growth and proliferation on nanofiber scaffold compared to cell culture in plastic. LSCs and MSCs successfully transfer from the nanofiber scaffold to the damaged ocular surface. Anti-inflammatory gene expression from nanofiber scaffold.	[187]
Scaffold-based corneal implant	Keratin/PVA	3D nanofiber structure prepared by applying the gas-foaming technique to electrospun nanofiber. Successful cell viability, infiltration, growth, and proliferation on 3D nanofiber. Compared to pristine nanofiber, 3D nanofiber shows improved transparency and mechanical properties. In in vivo studies, the 3D nanofiber shows improved biocompatibility in the rabbit eye compared to the pristine nanofiber.	[200]
Corneal wound dressing	COL/HA/PEO/GA/CS	Three different nanofiber formulations: (1) COL/HA/PEO electrospun nanofiber, (2) glutaraldehyde (GA) cross-linked form of electrospun nanofiber, and (3) CS-coated form of electrospun nanofiber. Compared to AM, CS-coated nanofiber has improved transparency and mechanical properties. Successful cell viability, proliferation, and biocompatibility in all nanofiber formulations. CS-coated nanofibers exhibit fibrosis-inhibiting properties similar to human AM. In in vivo studies, CS-coated and GA cross-linked nanofibers show re-epithelialization ability.	[188]
NIH3T3 fibroblast cell	PCLPCL/GEL	Two separate PCL and PCL/GEL nanofiber structures prepared by the electrospinning method. PCL/GEL nanofiber is more hydrophilic than PCL nanofiber. Cells on random and aligned PCL/GEL nanofibers have a better growth rate than PCL nanofibers. The orientation of the nanofiber matrix does not affect cell adhesion and proliferation, while the cells on the aligned nanofiber elongate parallel to the fibers. Fibroblast cells on aligned nanofibers express genes associated with actin production, actin polymerization, and focal adhesion formation.	[201]
The lipid phosphate phosphatase-related	E-PA	Nanofiber structure produced by the self-assembly method. Biocompatible nanofibers. In vitro studies, LPPR–PA nanofibers inhibit VEGF-induced cellular migration and proliferation. LPPR–PA nanofibers exhibit a comparable suppressive effect to bevacizumab in the in vitro angiogenesis assay. In in vivo studies in rat eyes, LPPR–PA nanofibers reduce corneal neovascularization more effectively than bevacizumab.	[202]

COL: collagen, CS: chitosan, dAM: decellularized amniotic membrane, DPG: _D_-phenylalanine gelators, E-PA: P Lauryl-VVAGE-Am, GA: glutaraldehyde, GEL: gelatin, HA: hyaluronate (HA), LPG: _L_-phenylalanine gelators, PA: peptide amphiphile, PA6/12: polyamide 6/12 (PA6/12), PCL: polycaprolactone, PEO: polyethylene oxide, PHBV: poly (3-hydroxybutyrate-co-3-hydroxyvalerate), PLA: poly(lactic acid), PLCL: poly(L-lactic acid-co-3-caprolactone), PLGA: poly(lactic-co-glycolic acid), PPy-G: polypyrrole functionalized graphene, PVA: polyvinyl alcohol RPG: a racemic mixture of LPG and DPG, and SF: silk fibroin.

## 4. Summary

Today, nanofiber technology can be used in many fields, from textiles to sensor production, from drug delivery systems to tissue engineering. Although there are different production methods, electrospinning is the most widely used among these methods. The most important reasons for this are the ease of controlling the fiber morphology, its versatility, and its ability to produce high-quality fiber. In addition, one of its unique advantages is the ability to prepare nanofibers with different properties by using different tools and techniques in the electrospinning method.

As a drug delivery system, nanofibers can be targeted to different tissues in the treatment of many diseases. Ocular applications are one of the most prominent of these areas. Nanofiber-based ocular implants or inserts are frequently studied as drug delivery systems. The high biocompatibility of these systems and the apparent long drug release times are promising in terms of increasing patient compliance and bioavailability in ocular drug administration.

Due to their large surface area, highly porous structure, preparation with different polymers, high drug loading capacity, and special morphologies, nanofibers attract a lot of attention not only as drug delivery systems but also in tissue engineering. Tissue engineering studies are important, especially in ocular tissue destruction, which is frequently encountered due to environmental factors.

In recent years, in particular, positive and promising results from ocular drug delivery systems and tissue engineering studies with nanofibers suggest that nanofiber research will become even more popular in the upcoming years. With the development of the current scale-up studies, nanofibers also offer a tremendous alternative to the conventional drug carrier systems and biomaterials used in ocular targeting.

## Figures and Tables

**Figure 1 pharmaceutics-15-01062-f001:**
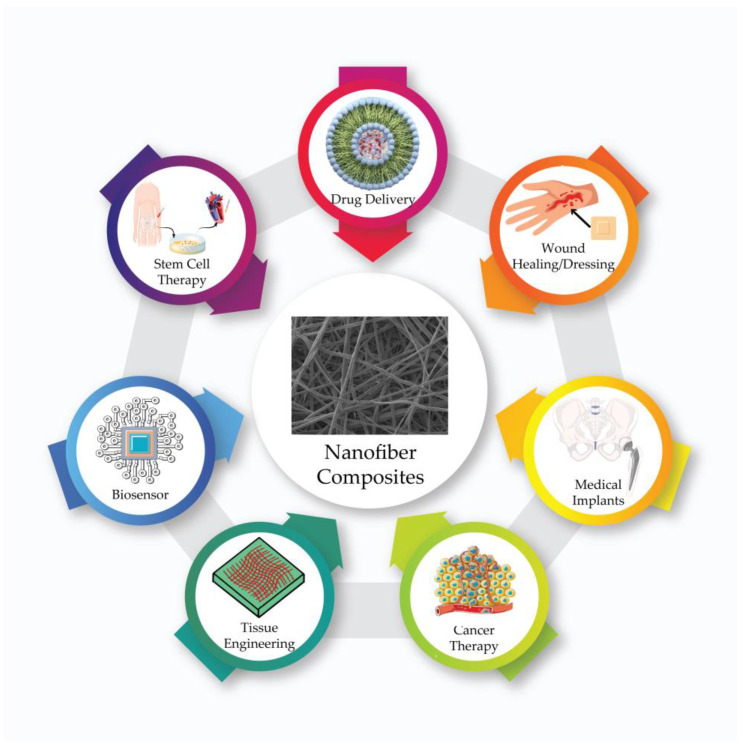
Biomedical applications of nanofiber composites.

**Figure 2 pharmaceutics-15-01062-f002:**
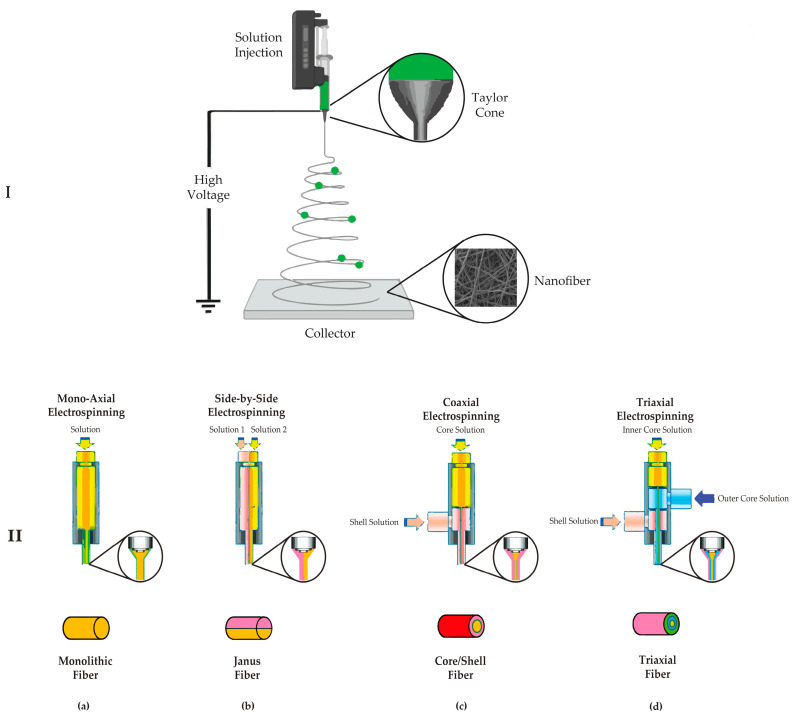
(**I**) Schematic diagram of the electrospinning apparatus and Taylor’s cone. (**II**) Schematic examples of electrospinning spinnerets: (**a**) monoaxial electrospinning, (**b**) side-by-side electrospinning, (**c**) coaxial electrospinning, and (**d**) triaxial electrospinning.

**Table 1 pharmaceutics-15-01062-t001:** List of parameters affecting properties of electrospun nanofibers.

	Parameter	Effect
Processing Parameters	Voltage	As the voltage increases, the diameter of the fiber generally decreases.
	Flow rate	A higher flow rate increases the diameter of the fiber and can cause bead formation. The flow rate is usually a maximum of 1 mL/h.
	Tip-to-collector distance (TCD)	Longer distance results in finer fibers. If the distance is too short, a nonuniform beaded film is formed. The diameter of the nanofiber increases with a decrease in TCD.
Solution Parameters	Concentration of solution	A higher concentration increases the nanofiber diameter and reduces the chance of bead formation. However, excessively high concentrations may clog the nozzle. A low concentration may cause sputtering.
	Viscosity	As the viscosity increases, coarser and continuous nanofibers are formed, while low-viscosity solutions cause the formation of finer and shorter nanofibers.
Solvent Parameters	Volatility of solvent	As the volatility of the solvent increases, a more porous and large surface area is formed.
	Dielectric constant	If the dielectric constant of the solution is low, beaded fibers may form. As the dielectric constant increases, the nanofiber diameter decreases.
Ambient Parameters	Temperature	Temperature changes viscosity and the rate of solvent evaporation. Generally, higher temperatures result in lower viscosity and more effective solvent evaporation.
	Humidity	If the humidity is too high, bead and porous structures may occur on the nanofiber.

**Table 2 pharmaceutics-15-01062-t002:** Examples of synthetic and natural polymers used for the creation of nanofibers for drug delivery purposes and their solvents.

**Polymer**	**Solvent**	**Polymer Physicochemical Properties**	**Polymer Properties**	**Polymer Type**	**References**
CA	DMF:ACN	Hydrophobic	Nontoxic, nonirritant, and biodegradable.	Natural	[47]
Chitosan	TFA:DCM/AA	Hydrophilic	Nontoxic, mucoadhesive, and biodegradable.	Natural	[48,49]
Collagen	HFIP	Hydrophilic	Biodegradable and nontoxic.	Natural	[50]
Gelatin	AA	Hydrophilic	Biocompatible and biodegradable.	Natural	[51]
PAN	DMF	Hydrophobic	Mechanically and thermally stable, low density.	Synthetic	[52]
PCL	DCM:DMF	Hydrophobic	Biocompatible and has a wide range of molecular weight.	Synthetic	[53]
PEG	Deionized water: ethanol	Hydrophilic	Nontoxic, inert, and biocompatible.	Synthetic	[54]
PEO	Ethanol/Deionized water	Hydrophilic	Biocompatible, biodegradable, and good conductivity.	Synthetic	[55]
PLA	DCM	Hydrophobic	Biodegradable and slow degradation rate.	Synthetic	[56]
PLGA	DCM:DMF or ACE:EtAc	Hydrophobic	Biocompatible, biodegradable, and has adjustable mechanical characteristics.	Synthetic	[57]
PMMA	DMF	Hydrophobic	Biocompatible and good conductivity.	Synthetic	[58]
PU	HFIP	Hydrophilic	Good conductivity.	Synthetic	[59]
PVA	Deionized water	Hydrophilic	Nontoxic, amorphous, and temperature and polymerization degree-dependent solubility.	Synthetic	[60]
PVP	10% ethanol Deionized water	Hydrophilic	Nontoxic, temperature-resistant, pH-stable, biocompatible, and biodegradable.	Synthetic	[61]
SF	Formic acid	Hydrophilic	Biocompatible and biodegradable.	Natural	[62]

AA: acetic acid; ACE: acetone; CA: cellulose acetate; DCM: dichloromethane; DMF: dimethylformamide; EtAc: ethyl acetate; HFIP: hexafluoro-2-propanol; PAN: polyacrylonitrile; PCL: polycaprolactone; PEG: poly(ethylene glycol); PEO: polyethylene oxide; PLA: polylactic acid; PLGA: poly lactic-co-glycolic acid; PMMA: polymethyl methacrylate; PU: polyurethane; PVA: polyvinyl alcohol; PVP: polyvinylpyrrolidone; SF: silk fibroin; TFA: trifluoroacetic acid.

**Table 3 pharmaceutics-15-01062-t003:** Different types of electrospinning and their advantages and disadvantages.

Electrospinning Type	Advantages	Disadvantages	References
Blend electrospinning	Simple process with a wide range of polymer combinations. Produces homogeneous fibers with improved properties. Enables the combination of different functionalities and applications in a single fiber.	Limited control over the fiber morphology and composition compared to other methods. Potential for phase separation or incompatibility. Limited control over the distribution of different polymers in the fiber.	[64,65]
Coaxial electrospinning	Produces core-sheath or hollow fibers with controlled dimensions and properties. Ability to encapsulate sensitive drugs and bioactive molecules. Controlled release of the core material.	More complex setup and operation compared to other electrospinning methods. Prone to clogging and instability issues. Difficult to produce fibers with a uniform diameter.	[66]
Emulsion electrospinning	Enables the encapsulation of hydrophilic or hydrophobic agents within the fibers. Offers controlled release profiles and improved stability of the encapsulated agents. Can produce fibers with different morphologies, such as beads-on-string or Janus fibers.	More complex process compared to blend electrospinning methods. Requires the use of surfactants or stabilizers to form the emulsion. Difficult to control droplet size and distribution.	[64]
Melt electrospinning	The ability to process a wide range of materials including thermoplastics, elastomers, and composites. Offers high production rates. Produces fibers with improved mechanical properties and thermal stability.	Prone to degradation or crosslinking of the polymer during the processing. Limited choice of materials due to high melting points. Increased risk of polymer degradation.	[64,65,67]
Gas-jet electrospinning	A simple process that does not require a high-voltage power supply. Enables the production of fibers with controlled dimensions and alignment. Can be used to fabricate hybrid fibers with different materials, such as metals or ceramics. Ability to process materials with high viscosity.	Limited production rates compared to other electrospinning methods. Requires high gas flow rates, which can be expensive.	[68]
Side-by-side electrospinning	Produces fibers with tailored compositions, properties, and functionalities. Enables the fabrication of complex fiber structures, such as core-shell or Janus fibers. Controlled release of the two materials.	Requires precise control over the two polymer phases. Potential for phase separation or incompatibility. Limited control over the interface between the two polymers.	[69]

**Table 4 pharmaceutics-15-01062-t004:** Nanofiber-based ocular drug delivery systems.

Loaded Drug	Polymer	Comments	References
Amphotericin B	PLGA/Eu-L/Gellan Gum/ Pullulan, Eu-L/Gellan Gum/Pulluln	Homogeneous nanofibers containing both nanoparticle and polyelectrolyte complexes separately. A nanoparticle-based system that transforms into nanofiber in situ. Good cellular tolerance in all formulations. Higher antifungal effect of polyelectrolyte complex nanofiber compared to other formulations.	[153]
Azithromycin	PLGA/PL/PVP	Successfully produced azithromycin-loaded nanoparticle-in-nanofiber inserts. Homogeneously dispersed nanoparticles in the nanofiber. Inserts show sustained drug release over 10 days in in vitro drug release tests.Biodegradable and biocompatible inserts. As a result of in vivo pharmacokinetic studies, inserts have a 14.8-fold higher AUC_t_ compared to eye drops. As a result of in vivo pharmacokinetic studies, inserts have a 1.6-fold higher C_max_ compared to eye drops.	[154]
Besifloxacin	HP-β-CD/PLC/PEG	Drug loading efficiency is over 90% in all formulations. Not high cytotoxicity in any of the formulations compared to the control product. Burst drug release in the first 2 days followed by a slow-release profile. In ex vivo drug permeability studies, the drug delivery of nanofibers was close to the commercial drug. In in vivo studies, infected corneas were significantly treated.	[155]
Bevacizumab	PVA/PCL/Gelatin	Nanofibers with the appropriate release profile for all samples, based on the cumulative amount of drug released in 6 days. The release rate of coaxial bevacizumab nanofibers is lower than that of PVA-only bevacizumab nanofibers. In the CAM model experiment, all bevacizumab nanofibers showed antiangiogenic activity. All formulations are nontoxic.	[156]
Brimonidine tartrate	PAMAM-mPEG/PEO	Successfully produced electrospun dendrimer-based nanofibers. Slower drug release from nanofibers compared to dendrimer and solution forms. Compared to the solution form, the nanofibers have no difference in permeability. Equivalent drug efficacy between nanofibers and solution form in terms of single dose IOP response in in vivo test results. However, with continuous long-term (3 weeks) use of nanofibers, there is a significant decrease in the average intraocular pressure (IOP) values compared to the solution form. Fast-dissolving, biodegradable, and biocompatible fibers.	[157]
Brinzolamide	β-CD/HPC/PCL	Possibility of more accurate dosing compared to control eye drops. Near-linear drug migration through ex vivo sheep corneas over 6 h reaching therapeutic concentration in the receptor medium. All formulations are biocompatible.	[158]
Cyclosporine A (CsA)	PLA	According to in vivo test results, CsA-loaded nanofibers significantly decreased the number of CD3-positive cells (T lymphocytes) and the production of proinflammatory cytokines in the corneas compared to eye drops and placebo. Corneal inflammation and corneal neovascularization are effectively suppressed by CsA-loaded nanofibers. Central corneal thickness returned to preinjury levels only in corneas treated with CsA-loaded nanofibers. More sustained drug release than eye drops, despite lower drug concentration.	[159]
Dexamethasone	PLA/PVA	Drug release profiles of electrospun nanofiber inserts are more stable than solvent-cast polymeric inserts. The cytotoxicity of electrospun nanofiber inserts is less than solvent-cast polymeric inserts. The thickness of electrospun nanofiber inserts is less than solvent-cast polymeric inserts. The electrospinning method is more suitable than the solvent casting method for preparing inserts with PLA/PVA polymers.	[160]
	Succinic anhydride	Nanofibrous hydrogel structure produced by the self-assembly method. Sustained drug release is influenced by the initial pH of the hydrogel. Biodegradable and biocompatible. Higher precorneal retention compared to the solution form. Higher ocular bioavailability compared to the solution form.	[161]
Dexamethasone gentamicin	PVP/KP188	Nanofibers have a highly porous structure and a great surface-to-volume ratio. Nanofibers dissolve rapidly in the tear fluid upon contact with the ocular surface. Over 92% drug loading for both active ingredients. Biocompatible formulation. In the ex vivo microfluidic cornea model, the nanofiber insert has a much longer residence time compared to fluid eye drops. Compared to eye drops, the AUC_20–60min_ increased by 342.54%.	[162]
Dexamethasone acetate	PCL	Burst drug release of approximately 47% in the first 2 days. Complete degradation and 100% drug release in 12 days. Successful cellular biocompatibility on ARPE-19 and MIO-M1 cells in in vitro tests. Excellent biocompatibility after vitreous implantation in mouse eye in in vivo tests.	
Dorzolamide	PLGA/PEG/PVA	Nanoparticles prepared by freeze-milling of electrospun nanofibers. 2 times higher AUC compared to commercial eye drops in in vivo tests. 2 times higher duration of IOP reduction compared to commercial eye drops in in vivo tests. Enhanced preocular retention. Safe and biocompatible formulation.	[163]
Doxorubicin	Glycopeptide	A nanoparticle-based system that transforms into nanofiber in situ. Complete restoration of physiological angiogenesis and reduced pathological neovascularization in the mouse model of oxygen-induced retinopathy (OIR). Having good histocompatibility. Long-term accumulation within cells within 24 h, but significant reduction in the intracellular drug when incubation period exceeds 6 h.	[164]
ε-polylysine ferulic acid	PVP/HA	Electrospun nanofiber inserts loaded with ferulic acid and cross-linked with ε-polylysine successfully produced. Ferulic acid completely released from the inserts in 20 min. ε-polylysine completely released from the inserts in 30 min. Antimicrobial activity against *Pseudomonas aeruginosa* and *Staphylococcus aureus*. Biocompatible and nonirritant inserts as a result of Hen’s Egg Test Chorioallantoic Membrane (HET-CAM) assay.	[31]
Fluocinolone acetonide	PCL	Drug loading efficiency is over 95%. Sterile, nonirritating, and biocompatible nanofiber inserts. Slow biodegradable nanofiber inserts. Extended drug release and higher retention time compared to commercial eye drop drugs. Higher t_1/2_ and AUC values, hence better bioavailability compared to commercial eye drop drugs in in vivo pharmacokinetic studies.	[165]
Forskolin	SA/PVA	Drug encapsulation of nanofibers is between 94.70 ± 0.26% and 96.90 ± 0.58%. According to the Higuchi equation, nanofibers produce zero-order kinetic controlled drug release. In in vivo intraocular pressure (IOP) reduction studies, nanofibers provided a significant and controlled reduction in IOP for up to 45 h.	[166]
Itraconazole	CA/PVA/PCL/PEG	Antifungal activity against candida albicans and Aspergillus fumigatus in all formulations. Approximately 50–70% of the drug is released over 55 days (prolonged drug release). Cell viability > 70% at all drug concentrations. Safe, non-irritant, and suitable nanofibers.	[167]
Levofloxacin	PLA	Nanofiber in the form of composite scaffolds. Drug releasing 50% of the levofloxacin content with burst effect on the first day, and between 2–7 days sustained drug releasing approximately 90% of the total levofloxacin content. Despite low levofloxacin content, excellent therapeutic effects in in vivo rabbit models by promoting structural and functional restoration of conjunctiva after transplant.	[168]
	PCL	Drug-loaded nanofiber sutures prepared by electrospinning method. Nanofiber sutures conform to U.S.P. specifications in terms of size and strength. 96% retained breaking strength over 31 days. Drug release detected in rat eyes for at least 30 days. Biodegradable and biocompatible sutures. More effective ocular bacterial infection prevention for 1 week compared to drug solution in in vivo studies.	[169]
Moxifloxacin hydrochloride pirfenidone	PVP/PLGA	The nanofibers showed an antimicrobial effect for 24 h in the zone of inhibition test. The antimicrobial and anti-scarring properties of the two drug-loaded nanofibers were found to be substantially equivalent to the free solutions of the two drugs in the Western blot method. The AUC_0–24h_ of moxifloxacin HCl and pirfenidone in nanofibers is 1.77 times and 2.49 times higher than in solution form, respectively. The t_1/2_ of moxifloxacin HCl and pirfenidone in nanofibers is 2.34 times and 1.43 times higher than in solution form, respectively.	[170]
Ofloxacin	CS/PVA/Eu-RL/GA	Four different nanofiber formulations: (1) single-layered nanofibers with PVA/CS polymers, (2) multilayered nanofibers with Eu-RL polymers as the outer layer and PVA/CS polymers as the inner layer, (3) GA cross-linked formulation of single-layered nanofiber, and (4) GA cross-linked formulation of multilayered nanofiber. All formulations have a drug loading capacity above 95%.Crosslinking of nanofibers causes an increase in their fiber diameters. In the inhibition zone test, single-layered nanofibers have a greater antimicrobial effect than multilayered nanofibers. Crosslinking reduces burst drug release in nanofibers. Enhanced prolonged drug release in multilayered nanofibers. Higher AUC in all formulations compared to the solution form of the drug. Cross-linked multilayered nanofiber has the highest mean residence time (MRT). All formulations are nonirritant and biocompatible.	[171]
Silver (Ag) nanoparticles	CNF/PLA	Nanofiber membranes coated with CNF containing silver nanoparticles homogeneously on the surface. High ocular biocompatibility and cell proliferation in in vitro cell culture experiments. Provides over 95% inhibition of *Escherichia coli* (*E. coli*) and *Staphylococcus aureus* (*S. aureus*). Approximately 75% antifungal effect against Fusarium spp.	[172]
Triamcinolone acetonide	Zein/Eu-S PVP/CS PVA/CS PVP/PVA/CS	The aim is to compare the properties of nanofibers produced from different polymers. Best quality, smallest diameter (120 ± 30 nm), and homogeneous structure: PVP/CS electrospun nanofiber. As a result of in vitro drug release tests, the same formulation (PVP/CS) is the only nanofiber that follows the zero-order kinetic profile. All formulations show a prolonged drug release profile.	[173]
Timolol maleate	PVP/PNIPAM	Novel drug delivery systems are designed as contact lenses coated with nanofibers. Drug release is based on quasi-Fickian or Fickian diffusion with the application of Higuchi and Korsmeyer–Peppas models. Biological evaluation using freshly cut bovine cornea confirms the biocompatibility of the formulations. Nanofibers containing permeation enhancers (borneol) release 18.47% more drugs than nanofibers without penetration enhancers (86.71% and 68.24% drug releases, respectively).	[174]
	PVA/PL	Successfully prepared in situ gelling nanofiber system. Drug loading efficiency is over 98.8%. 100% drug release with first-order kinetics in 15 min.Increased ex vivo permeability compared to the solution form. Higher drug retention in the corneal tissue. Sustained IOP lowering effect for up to 24 h compared to drug solution in in vivo tests.	[175]
	Ac-(RADA)4-CONH2 peptide solution Ac-(IEIK)3I-CONH2 peptide solution	Nanofibrous hydrogel structures produced by the self-assembly method. The aim is to compare the properties of hydrogel nanofibers produced from different peptide solutions. Ac-(IEIK)3I-CONH2 hydrogel shows slower drug release. Slower ex vivo penetration for Ac-(IEIK)3I-CONH2 hydrogel due to the drug release rate. Biocompatibility and safety for both hydrogels as a result of histological tests. Enhanced bioavailability and efficient reduced IOP results up to 24 h for Ac-(RADA)4-CONH2 hydrogel in in vivo pharmacodynamic and pharmacokinetic tests.	[176]
Timolol maleate Brimonidine tartrate	Ac-(RADA)4-CONH2 peptide solution	Two separate APIs encapsulated nanofiber hydrogel structures produced by the self-assembly method. Burst drug release of approximately 60% in the first 1 h for both APIs. Drug release over 8 h in total. Enhanced ex vivo permeability for both drugs compared to solution forms. No significant change in the structural integrity of the corneas.	[177]
Vitamin C Zinc (Zn)	LDH/ PUU	The drug release of nanofibers (LDH/PUU) is prolonged from 5 h to 5 days compared to nanoparticles (LDH). Biocompatible nanofibers. All scaffolds have homogeneous nanofiber morphology.	[178]

β-CD: β-cyclodextrin, HP-β-CD: BH-hydroxypropyl-beta-cyclodextrin, CA: cellulose acetate, CNF: cellulose nanofibrils, CS: chitosan, Eu-L: Eudragit L, EU-RL: Eudragit RL-100, EU-S: Eudragit S100, GA: glutaraldehyde, HA: hyaluronic acid, HPC: hydroxypropyl cellulose, KP188: Kolliphor P188, LDH: Zn−Al-layered double hydroxide, mPEG: methoxy polyethylene glycol, PAMAM: polyamidoamine, PEG: polyethylene glycol, PCL: polycaprolactone, PEO: polyethylene oxide, PL: Pluronic F-127 (Poloxamer 407), PLA: poly(lactic acid), PLGA: poly(lactic-co-glycolic acid), PNIPAM: poly (N-isopropylacrylamide), PUU: poly(urethane-urea), PVA: polyvinyl alcohol, PVP: polyvinylpyrrolidone, and SA: sodium alginate.

## Data Availability

Not applicable.

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
