# Peer review of "Nanofibers in Ocular Drug Targeting and Tissue Engineering: Their Importance, Advantages, Advances, and Future Perspectives"

_pharmaceutics, 2023, doi:10.3390/pharmaceutics15041062_

Round 1

Reviewer 1 Report

The review is focused on introducing the different modes of nanofiber fabrication and highlighting the applications pertaining to ocular drug delivery and tissue engineering. The manuscript is well-written and will appeal to a wide range of readers. However, there are some major concerns that need to be addressed to improve the quality of the review. Especially in terms of providing thoughts from the authors’ perspective in terms of future directions and how the disadvantages can be addressed in further studies.

Major Concerns:

Lines 163-196 talking about the polymers is unnecessary at this stage as it deviates from the main topic. All these information can be summarized in table 2.

The different modes of electrospinning can be tabulated to provide the main advantages and disadvantages.

Provide the relevant references when stating a fact or providing a definition. Most of the references are provided at the end of each paragraph which makes it difficult to track the right citations. Additionally, the section talking about nanofibers for ocular systems lacks references.

It is unclear how the freeze-drying technique can be used to form nanofibers. It is more commonly used for producing three-dimensional scaffolds. Please provide more details on how this technique can be used for nanofiber mat production.

The authors bring up an important point- “Although the drug has been proven to increase penetration and permeation with ocular tissues, they can cause blurred vision, irritation, and watery discharge.” At the end of this section, please expand upon what are some potential ways that this can be avoided moving forward. The current version of the manuscript only highlights what is existing in the literature, so adding these perspectives based on the authors’ expertise would significantly improve the quality of the review. Also, please provide the same thoughts for the tissue engineering section.

There are several limitations to nanofiber fabrication using electrospinning such as the high cost, low yield, and requiring high voltages. Please highlight these drawbacks.

Another technique that needs to be highlighted is a fairly new technology called rotatory or centrifugal jet spinning (RJS/CJS). This technique has been studied widely in the last decade and has been shown to overcome several of the limitations associated with electrospinning. The authors are requested to review the following papers to add a section for this technique.

a)      Badrossamay, M. R., et al. "Engineering hybrid polymer-protein super-aligned nanofibers via rotary jet spinning." Biomaterials 35.10 (2014): 3188-3197.

b)     Loordhuswamy, A. M., et al. "Fabrication of highly aligned fibrous scaffolds for tissue regeneration by centrifugal spinning technology." Materials Science and Engineering: C 42 (2014): 799-807.

c)      Khang, A., et al. "Engineering anisotropic biphasic Janus‐type polymer nanofiber scaffold networks via centrifugal jet spinning." Journal of Biomedical Materials Research Part B: Applied Biomaterials 105.8 (2017): 2455-2464.

d)     Ren, L., et al. "Highly efficient fabrication of polymer nanofiber assembly by centrifugal jet spinning: process and characterization." Macromolecules 48.8 (2015): 2593-2602.

e)     Gonzalez, GM., et al. "Production of synthetic, para‐aramid and biopolymer nanofibers by immersion rotary jet‐spinning." Macromolecular Materials and Engineering 302.1 (2017): 1600365.

f)       Ravishankar, P., et al. "Using dimensionless numbers to predict centrifugal jet-spun nanofiber morphology." Journal of Nanomaterials 2019 (2019).

g)      Bahú, Juliana O., et al. "Rotary Jet Spinning (RJS): A Key Process to Produce Biopolymeric Wound Dressings." Pharmaceutics 14.11 (2022): 2500.

Minor Concerns:

Please check for grammatical and typographical errors in the manuscript.

Figures 2 and 3 can be combined and discussed.

Please use the full terms when used for the first time. For instance, GFs in line 304. There are several of these such as SF, BMP-2, etc.

Crosscheck to ensure that terms are being used consistently. For instance, nanofibers vs nanofibres.

Check the bullet point numbering and organize the topics accordingly to improve readability.

There are several sentences pertaining to the electrospinning process characteristics in the introduction that is repetitive. Please reread the manuscript to highlight these at the beginning and redirect the readers to the introduction if there is a need to be repeated.

Author Response

We thank Reviewer for her/his valuable comments and suggestions that helped improve the quality of our manuscript. The following responses were prepared to address all of the reviewers’ comments in a point-by-point manner.

General Comments: The review is focused on introducing the different modes of nanofiber fabrication and highlighting the applications pertaining to ocular drug delivery and tissue engineering. The manuscript is well-written and will appeal to a wide range of readers. However, there are some major concerns that need to be addressed to improve the quality of the review. Especially in terms of providing thoughts from the authors’ perspective in terms of future directions and how the disadvantages can be addressed in further studies.

Response: Thank you for your comment and encouragement that our manuscript is promising. We have revised the manuscript taking into account your comments. We've shared information about all the changes we've made in the specific comments.

Specific Comments:

Comment 1.

Lines 163-196 talking about the polymers is unnecessary at this stage as it deviates from the main topic. All these information can be summarized in Table 2.

Response 1.

In line with your comment, we have updated the polymer table and removed the relevant sections from the text to avoid duplication of information.

Comment 2.

The different modes of electrospinning can be tabulated to provide the main advantages and disadvantages.

Response 2.

Thank you for your valuable comment, which will strengthen the content of the nanofiber section of the manuscript. Under the "types of electrospinning" section on line 292, we added a table with the name "Table 3. Different types of electrospinning and their advantages and disadvantages" comparing all methods.

Comment 3.

Provide the relevant references when stating a fact or providing a definition. Most of the references are provided at the end of each paragraph which makes it difficult to track the right citations. Additionally, the section talking about nanofibers for ocular systems lacks references.

Response 3.

Thank you for your comments. In line with your comment, we have cited all of the references within the paragraphs to show from which reference the quotations were made. Please accept our apologies for the situation with the ocular department. We rearranged this section.

Comments 4.

It is unclear how the freeze-drying technique can be used to form nanofibers. It is more commonly used for producing three-dimensional scaffolds. Please provide more details on how this technique can be used for nanofiber mat production.

Response 4.

Thank you for your valuable comment, which will strengthen the content of the nanofiber section of the manuscript. We have made the freeze-drying technique section more comprehensive and explanatory with new references. Please find this section between lines 411-448.

Comments 5.

The authors bring up an important point- “Although the drug has been proven to increase penetration and permeation with ocular tissues, they can cause blurred vision, irritation, and watery discharge.” At the end of this section, please expand upon what are some potential ways that this can be avoided moving forward. The current version of the manuscript only highlights what is existing in the literature, so adding these perspectives based on the authors’ expertise would significantly improve the quality of the review. Also, please provide the same thoughts for the tissue engineering section.

Response 5.

Thank you for your comment. By making changes in the section related to ocular drug delivery systems, we have included both our literature knowledge and our comments based on our experience. We have highlighted the advantages and disadvantages of nanofibers over other carrier systems, emphasizing how existing disadvantages can be avoided.

Comments 6.

There are several limitations to nanofiber fabrication using electrospinning such as the high cost, low yield, and requiring high voltages. Please highlight these drawbacks.

Response 6.

Thank you for your comment. We discussed the limiting points of electrospinning with new references. Please find this section between 125-133.

Comments 7.

Another technique that needs to be highlighted is a fairly new technology called rotatory or centrifugal jet spinning (RJS/CJS). This technique has been studied widely in the last decade and has been shown to overcome several of the limitations associated with electrospinning. The authors are requested to review the following papers to add a section for this technique.

Response 7.

Thank you for your valuable comment, which will strengthen the content of the nanofiber section of the manuscript. In line with your comment, we have added a section on rotatory or centrifugal jet spinning (RJS/CJS). Please find this section between 449-512.

Comments 8.

Please check for grammatical and typographical errors in the manuscript.

Response 8.

We are sorry about our typographical errors and grammatical mistakes. The language revision and edition of the text were conducted according to your warnings. While making the revisions, we used the Grammarly program, which many schools and newspapers refer to. If you have any problems related to language revision, please let us know.

Comments 9.

Figures 2 and 3 can be combined and discussed.

Response 9.

Thank you for your comment. We combined figures 2 and 3.

Comments 10.

Please use the full terms when used for the first time. For instance, GFs in line 304. There are several of these such as SF, BMP-2, etc.

Response 10.

We apologize for the missing information provided. We've checked all the abbreviations in the manuscript and are more descriptive by putting their full names where they first appear.

Comments 11.

Crosscheck to ensure that terms are being used consistently. For instance, nanofibers vs nanofibres.

Response 11.

Sorry for the spelling differences. We have ensured that all terminology in the Manuscript is similar.

Comments 12.

Check the bullet point numbering and organize the topics accordingly to improve readability.

Response 12.

Thank you for your comment. The substance marking and numbering system is written in the order specified by the Pharmaceutics journal. Unfortunately, we could not make any changes here as it would be against the rules of the journal. However, if the editors approve, we can make the changes.

Comments 13.

There are several sentences pertaining to the electrospinning process characteristics in the introduction that is repetitive. Please reread the manuscript to highlight these at the beginning and redirect the readers to the introduction if there is a need to be repeated.

Response 13.

Thank you for your comment. Repeated sentences about the electrospinning process characteristics in the manuscript were excluded and all information about it was given in the introduction part.

Reviewer 2 Report

The title of this paper points out the advantages, importance, and advances of nanofibers in tissue engineering for targeted drug delivery in the eye, and the lack of comparison with other commonly used methods in the content makes it difficult to reflect the advantages. It may be appropriate to add the comparison of current commonly used methods to highlight its unique advantages.

Table 1 shows that the higher concentration increases the nanofiber diameter, the higher and the higher viscosity increases the thickness of the nanofiber. Is there any difference between the diameter and the thickness?

The focus of this article needs to be clarified. The title focuses on the application of Nanofibers in Ocular Drug Targeting. In the previous section, various preparation methods of nanofibers and different material characteristics are difficult to fit with the proposed eye engineering. In turn, it is better to explain the characteristics of the eye environment and the required materials. Then it can be shown that the nanofibers conform to the requirements.

The word “uniform nanofibers” in table 3 is recommended to use the word “homogeneous nanofibers”

Author Response

We thank Reviewer for her/his valuable comments and suggestions that helped improve the quality of our manuscript. The following responses were prepared to address all of the reviewers’ comments in a point-by-point manner.

Comment 1.

The title of this paper points out the advantages, importance, and advances of nanofibers in tissue engineering for targeted drug delivery in the eye, and the lack of comparison with other commonly used methods in the content makes it difficult to reflect the advantages. It may be appropriate to add the comparison of current commonly used methods to highlight its unique advantages.

Response 1.

Thank you for your comment. Required comparison with the commonly used methods was included. In the section on drug carrier systems, we have also explained in detail the advantages of nanofibers over other carrier systems. In the tissue engineering section, we tried to emphasize the importance of nanofibers by emphasizing the similarities of nanofibers with ocular tissues.

Comment 2.

Table 1 shows that the higher concentration increases the nanofiber diameter, the higher and the higher viscosity increases the thickness of the nanofiber. Is there any difference between the diameter and the thickness?

Response 2.

Thank you for your comment. No, there is no difference. It was meant to say that the diameter of the nanofiber (thickness) increases at higher concentrations and viscosity conditions.

Comment 3.

The focus of this article needs to be clarified. The title focuses on the application of Nanofibers in Ocular Drug Targeting. In the previous section, various preparation methods of nanofibers and different material characteristics are difficult to fit with the proposed eye engineering. In turn, it is better to explain the characteristics of the eye environment and the required materials. Then it can be shown that the nanofibers conform to the requirements.

Response 3.

Thank you for your valuable comment. Since our article was designed to be featured in a special issue titled "Advances in Ocular Drug Delivery" under the main title of "Drug Delivery and Controlled Release", it was our primary goal to highlight nanofibers. For this reason, we first mentioned nanofibers. However, in line with your comment, we have added the "3.1. Anatomy and Physiological Barriers of Eye" subtitle under the "3.Nanofibers as Ocular System" main title. In this section, we explained on the anatomy, physiology, and barriers of the eye.

Comments 4.

The word “uniform nanofibers” in table 3 is recommended to use the word “homogeneous nanofibers”

Response 4.

Thank you for your comment. We changed all “uniform nanofibers” to “homogeneous nanofibers”.

Reviewer 3 Report

The present manuscript is a review study that focuses on using nanofibrous layers for applications in ophthalmology. Approximately one-half of the text is devoted to general information concerning the preparation of such materials, gradually drawing the reader into the narrow subject matter in a readable manner. The other half is already devoted to the targeted field, here in the form of tables with numerous references and comments by the authors of the study. The contribution of the study and the interest of the readers can be assessed at a high level, therefore the presented text can be recommended for publication after supplementation and modification in the following points:

Please go ahead and follow the format of paragraph numbering throughout the text, which ends at level 2.1.1. Then, on line 334, the full chapter title is missing.

The paragraph describing the Gas Jet Electrospinning method starts on line 265. The vast majority of upgrades use only airflow, and only units of publications also use heated air. But the text here implies that this method always uses heated air. Yet the significant benefit is the airflow itself. Furthermore, "... heat near the nozzle and delay..." should be correct because it does not delay but accelerates solidification. The following sentence starting on line 268 makes no sense. The text then goes on to discuss the method of filamentation from the melt, which is different and described in the previous paragraph. For this reason, the text is a little unintelligible and confusing. I recommend that you consider correcting this paragraph.

Paragraphs 3.1 and 3.2 consist mainly of tables of individual references. I recommend adding a summary at the end of each paragraph to capture the significant findings from the literature review. This could be, for example, trends that indicate future directions or also some interesting findings or issues that remain unresolved.

In the summary paragraph, on lines 509 and 510, there are assertions that do not follow the preceding text and are not sufficiently supported. Similarly, the sentence on line 525 refers to market research not mentioned in the preceding text. Please make corrections or additions to the text.

Author Response

We thank Reviewer for her/his valuable comments and suggestions that helped improve the quality of our manuscript. The following responses were prepared to address all of the reviewers’ comments in a point-by-point manner.

General Comments: The present manuscript is a review study that focuses on using nanofibrous layers for applications in ophthalmology. Approximately one-half of the text is devoted to general information concerning the preparation of such materials, gradually drawing the reader into the narrow subject matter in a readable manner. The other half is already devoted to the targeted field, here in the form of tables with numerous references and comments by the authors of the study. The contribution of the study and the interest of the readers can be assessed at a high level, therefore the presented text can be recommended for publication after supplementation and modification in the following points:

Response: Thank you for your comment and encouragement that our manuscript is promising. We have revised the manuscript taking into account your comments. We've shared information about all the changes we've made in the specific comments.

Specific Comments:

Comment 1.

Please go ahead and follow the format of paragraph numbering throughout the text, which ends at level 2.1.1. Then, on line 334, the full chapter title is missing.

Response 1.

Please accept our apologies. The title of this section has been added. It is "2.1.2. Non-electrospinning Methods" on line 339.

Comment 2.

The paragraph describing the Gas Jet Electrospinning method starts on line 265. The vast majority of upgrades use only airflow, and only units of publications also use heated air. But the text here implies that this method always uses heated air. Yet the significant benefit is the airflow itself. Furthermore, "... heat near the nozzle and delay..." should be correct because it does not delay but accelerates solidification. The following sentence starting on line 268 makes no sense. The text then goes on to discuss the method of filamentation from the melt, which is different and described in the previous paragraph. For this reason, the text is a little unintelligible and confusing. I recommend that you consider correcting this paragraph.

Response 2.

Thank you for your valuable comment, which will strengthen the content of the nanofiber section of the manuscript. We have rearranged the gas jet electrospinning section in line with your comment. Please find this section between lines 272-279.

Comment 3.

Paragraphs 3.1 and 3.2 consist mainly of tables of individual references. I recommend adding a summary at the end of each paragraph to capture the significant findings from the literature review. This could be, for example, trends that indicate future directions or also some interesting findings or issues that remain unresolved.

Response 3.

Thank you for your valuable comment. In line with your comment, we have added sections giving general information about the studies to these sections. In the section on drug carrier systems, we have also explained in detail the advantages of nanofibers over other carrier systems. In the tissue engineering section, we tried to emphasize the importance of nanofibers by emphasizing the similarities of nanofibers with ocular tissues.

Comments 4.

In the summary paragraph, on lines 509 and 510, there are assertions that do not follow the preceding text and are not sufficiently supported. Similarly, the sentence on line 525 refers to market research not mentioned in the preceding text. Please make corrections or additions to the text.

Response 4.

Thank you for your comment. We rearranged the summary section to make it compatible with the previous text.

Round 2

Reviewer 1 Report

The revised manuscript has addressed all the reviewer's comments to satisfaction and will attract a wide range of readership for the journal. 

Reviewer 2 Report

Accept